# Establishment of the Biotransformation of D-Allulose and D-Allose Systems in Full-Red Jujube Monosaccharides

**DOI:** 10.3390/plants12173084

**Published:** 2023-08-28

**Authors:** Fawei Liu, Shuangjiang Chen, Fuxu Pan, Zhihui Zhao, Mengjun Liu, Lili Wang

**Affiliations:** 1College of Horticulture, Hebei Agricultural University, Baoding 071001, China; lfw15731171935@163.com (F.L.); chsuji2@163.com (S.C.); pfx1234562022@163.com (F.P.); 2Research Center of Chinese Jujube, Hebei Agricultural University, Baoding 071001, China; lyzhihuizhao@126.com

**Keywords:** jujube juice, scarce sugar, double enzyme coupling, DPE, L-RI

## Abstract

In order to reduce sucrose content in jujube juice and prepare a jujube juice beverage rich in rare sugars, jujube juice was used as raw material for multienzyme catalysis in this study. The effects of single factors such as substrate, pH, DPE and L-RI addition ratio, enzyme treatment temperature, and metal ions on sucrose conversion and D-allulose formation in jujube juice were investigated. Changes in glucose, D-allulose, and D-allose contents in jujube juice before and after enzyme conversion were analyzed by high-performance liquid chromatography (HPLC). The results showed that ‘Xiangfenmuzao’ was more suitable for subsequent double enzyme coupling reactions in different varieties of jujube juice at different periods. Factors such as pH, DPE and L-RI enzyme ratio, temperature, and treatment time had significant effects on sucrose conversion and D-allulose production in ‘Xiangfenmuzao’ juice (*p* < 0.05). When the ratio of DPE and L-RI was 1:10, pH was 7.5, and the temperature was 60 °C for 7 h, the fructose content in the full-red stage jujube juice of ‘Xiangfenmuzao’ and ‘Jinsixiaozao’ decreased gradually, and the final yield was about 53%. The yield of D-allulose was about 29%, and the yield of D-allulose was about 17%. In this study, DPE and L-RI were used to treat whole red jujube juice, which could effectively reduce sucrose content in jujube juice and obtain a functional jujube juice beverage that is low in calories and rich in rare sugar.

## 1. Introduction

Jujube juice is a nutrient-rich beverage, containing cyclic adenosine monophosphate (cAMP), cyclic nucleotide monophosphate (cGMP), vitamin C, and polysaccharides [1,2,3,4,5,6]. It is a popular natural drink due to its numerous health benefits. Various varieties of jujube fruit are known for their sucrose accumulation, with fresh jujube containing up to 30% soluble sugars [7], including glucose, fructose, and sucrose, with sucrose being the most abundant [8]. However, as high sugar content in jujube juice can adversely impact the health of individuals with conditions such as diabetes, obesity, and cardiovascular diseases [9,10,11], converting the monosaccharides present in jujube juice into low-calorie natural rare sugars can facilitate the processing and use of jujube fruits. As people’s living standards continue to improve, they are becoming increasingly health conscious. Excessive sugar intake has a certain impact on human health, so there is an urgent need to find new types of sugar to replace traditional sweeteners. Rare sugars are a group of monosaccharides and their derivatives that occur in nature but in very small amounts [12] and were defined by the International Sugar Society (ISRS) in 2002 [13]. D-allulose has antioxidant activity and immune protection and is hypoglycemic, neuroprotective, and other properties [14,15,16,17,18,19]. D-allose has the functions of nerve protection, cancer treatment, cryoprotectant, strong antioxidant capacity, anti-hypertension, and so on [20,21,22,23,24,25,26,27]. Fifty types of rare sugars have been identified [12], which are classified into butyral, butanone, pentanal, pentanedose, hexanedose, and heptanedose according to different structures [12,28], while hexanedose is divided into D-type ketose and L-type ketose [29], such as D-allulose, D-tagatose, and L-sorbose, which are the most commonly studied rare sugars. Compared to traditional sweeteners, dilute sugars are comparable in sweetness but lower in calories, accounting for only 34% or less of traditional sweeteners, having higher stability tolerance and being non-hygroscopic, non-caries resistant, and highly tolerable, making up for the shortcomings of traditional sweeteners.

D-allulose can be synthesized by chemical synthesis [30,31] and biosynthesis [32,33]. However, chemical synthesis is a complex process that produces a large amount of waste and byproducts, which affects the taste of the final product. D-allulose is the C-3 isomer of D-fructose and is usually synthesized by D-psicose3-epimerase (DPE), which catalyzes the conversion of D-fructose. Most of the earliest heterologous expressions of D-psicose 3-epimerase in *E. coli*, which have a simple genetic background, are cost-effective, efficient, and have been used. DPE gene derived from *Clostridium* sp. BNL1100 was expressed in *E. coli* and purified by nickel column affinity chromatography to obtain DPE, which catalyzes the production of 120 g of D-allulose from 500 g of D-fructose [34]. Both immobilization techniques and whole-cell reactions can be used to express DPE heterologously using *E. coli* as the host bacterium for DPE [35]. D-allose is an aldose isomer of D-allulose and an epimer at the C-3 position of D-glucose. D-allose is produced by isomerizing D-allulose using L-rhamnose isomerase (L-RI). As the precursor of D-allose is D-allulose, and the production cost of D-allulose is high, D-glucose or D-fructose is generally used as the starting material in biosynthesis to produce D-allose through a series of conversion steps such as D-glucose isomerase (D-Glucose Isomerase), DTEase/DPEase, and L-RI.

At present, there have been reports on the conversion of DPE and L-RI to synthesize D-allulose and D-allose. When directly using the fructose substrate of the finished product to synthesize a single rare sugar, the synthesis conversion efficiency is not high, so waste is easily generated. In juice processing, glucose, fructose, and sucrose are directly used as substrates for conversion and synthesis, and there are few studies on the synthesis of rare sugars by DPE and L-RI. Based on the characteristics of high sucrose, glucose, and fructose content in jujube juice, sucrose was first hydrolyzed to glucose and fructose, and then glucose was converted to fructose. DPE and L-RI were used to convert fructose to D-allulose, and a small amount of D-aldose was converted to D-allulose. In this study, the optimal catalytic conditions for the coupling of DPE and L-RI double enzymes were studied to improve the conversion rate of D-allulose and D-allose in jujube juice. Prepare jujube juice into a nutritious juice drink containing D-allulose, D-allose and zero sucrose. The conversion of monosaccharides in jujube juice into low-calorie natural rare sugars can promote the processing of jujube fruits and extend the industrial chain of jujube processing, which provides a theoretical basis for the processing of jujube fruit into functional juice drinks.

## 2. Results

### 2.1. Analysis of Sugar Composition in Jujube Fruits of Different Varieties

The sugar composition of eight varieties of jujube at different times, including golden ‘Jinsixiaozao’, ‘Xiangfenmuzao’, ‘Huizao’, ‘Zanhuangdazao’, ‘Popozao’, ‘Linyilizao’, ‘Yazao’, and ‘Dasuanzao’, is shown in Figure 1. There were obvious differences in the sugar content of different varieties of jujubes. Except for ‘Xiangfenmuzao’, the sucrose content in the fruits of the other seven jujube varieties in the full-red stage accounted for a large proportion of the three monosaccharides, all of which were between 43.8–69.1%. ‘Xiangfenmuzao’, ‘Dasuanzao’, and ‘Yazao’ had a higher total sugar content. Among them, glucose and fructose accounted for 30% and 30.54% of the total sugar content, similar to the proportion of sucrose (31.26%). In the later experiments, glucose and fructose were mainly used, so ‘Xiangfenmuzao’ was more suitable for follow-up trials from the aspects of total sugar and monosaccharide content.

For each component, fructose, glucose, and sucrose content tended to increase with fruit development (except for large sour jujube), with glucose and fructose content varying significantly between the expansion stage and the white-ripening stage, and insignificantly between the semirubescent stage and the full-red stage (Figure 1A,B). Sucrose, on the other hand, began to accumulate in large quantities at the white-ripening stage, and the difference between white-mature stage and half-red stage was significant, while the difference between half-red stage and full-red stage was not significant (Figure 1C). The glucose and fructose content of big sour jujube showed an increasing trend from the young fruit stage to the semi-red stage and decreased from the semi-red stage to the full-red stage, whereas the sucrose content, on the contrary, decreased from the white-ripening stage to the semi-red stage and increased from the semi-red stage to the full-red stage. The proportion of fruit sugar components to total sugar varied between varieties at the full-red stage (Figure 1D). With the exception of ‘Xiangfenmuzao’, the sucrose content in the fruit of the other seven jujube varieties during the full-red stage accounted for a relatively large proportion of the three monosaccharides, all ranging from 43.8% to 69.1%, and belonging to the sucrose-accumulating type.

### 2.2. Induced Expression and Purification of D-Psicose-3-Epimerase and L-Rhamnose Isomerase

The recombinant plasmid DPE was transferred to *Bacillus subtilis* WB600, and positive colonies were added to LB-Kana liquid medium and cultured at 37 °C, 200 r/min. Recombinant plasmids were validated using primers. As shown in Figure 2A, PCR product band sizes ranged from 750 to 1000 bp by nucleic acid electrophoresis, which matched the expected results.

An expression of SDS-PAGE with a target protein size of 33 kDa was observed, consistent with the expected results. It can be seen from the figure that the DPE bands expressed in *Bacillus subtilis* are clear, and more than 90% of the purified pure target protein can be used in subsequent experiments (Figure 2B).

The recombinant plasmid L-RI was converted into *Escherichia coli* BL21 (DE3), and the root-picked positive colonies were transferred to LB-Amp liquid medium for 4 h at 37 °C and 200 r/min. Then, IPTG was added to induce its expression at 30 °C and 150 r/min. The recombinant plasmid L-RI gene fragment size was 1332 bp by nucleic acid electrophoresis, which was consistent with the expected results (Figure 2C).

The SDS-PAGE diagram of L-RI plasmid expression in *Escherichia coli*. The target protein size is 46.8 kDa, and it can be seen that the target protein band is brighter and the purification effect reaches 90%, which can be used for subsequent experiments (Figure 2D).

### 2.3. Effects of Different Reaction Conditions on D-Psicose-3-Epimerase and L-Rhamnose Isomerase Enzyme Activities

Temperature has a great influence on protein enzyme activity, and exceeding its optimal range will affect protein enzyme activity, and produce byproducts in industrial processing that have an impact on processing technology and its economic value. When increasing temperature, it can be observed that the activity of DPE and L-RI enzymes gradually increases, and the conversion efficiency of fructose increases in the same reaction time. At temperatures greater than 50 °C, the activity of DPE gradually decreases. At temperatures greater than 65 °C, the activity of L-RI gradually decreases. Therefore, the optimal reaction temperature for the DPE enzyme is 50 °C and the optimal reaction temperature for the L-RI enzyme is 65 °C (Figure 3).

Considering acidic conditions, as they change the spatial structure of enzyme proteins and the conformation of their enzymes. Environments that are too acidic or alkaline can affect enzymes. Therefore, it is necessary to study the optimal pH value of the enzyme to improve enzyme activity. The enzyme activity gradually increased as the pH of the reaction system increased. At pH 7.5, both DPE and L-RI enzyme activity was highest and higher than enzyme activity at other pH values (Figure 4). Therefore, the optimum reaction pH of DPE and L-RI is 7.5.

Different metal ions have different effects on the activity of DPE and L-RI enzymes, and some metal ions have inhibitory effects on the activity of DPE enzymes while some show promotional effects. In the absence of metal ion assistance, enzyme activity remains, but is weakened. Mn^2+^ had the strongest effect on DPE and L-RI enzyme activity (Figure 5). Compared with DPE enzyme activity without metal ions, which increased by 26.5%, L-EI enzyme activity increased by 38.1%, and Mn^2+^ was the most suitable metal ion for recombinant DPE and L-RI protein.

Mn^2+^ was known to have a strong promoting effect on DPE enzyme activity and L-RI enzyme activity. Different concentrations (0.5–3 mM) of Mn^2+^ were used to study their effects on DPE enzyme activity and L-RI enzyme activity (Figure 6). When 3 mM Mn^2+^ was added, the al effect on DPE enzyme activity and L-RI enzyme activity was strongest.

### 2.4. Kinetics of the Reaction of D-Psicose-3-Epimerase and L-Rhamnose Isomerase

Michaelis–Menten fitting was performed using GraphPad software to analyse the parameters Km and Vmax of the DPE enzyme and L-RI enzyme-catalyzed reaction. Km represents the substrate concentration when the enzyme-catalyzed reaction speed is half the maximum speed, and Vmax is the maximum enzyme-catalyzed reaction speed. When D-fructose was used as the substrate, the Km value of DPE was 25.82 mM and the Vmax was 6.0 pmol/min/mg (Figure 7a). L-RI has a Km of 15.5 mM and a Vmax of 2.3 pmol/min/mg (Figure 7b).

### 2.5. Study of Catalytic Conditions for DPE and L-RI Double Enzyme Coupling

The difference between the optimal temperature and 75 °C double enzyme coupling for the synthesis of D-allose and D-allulose was significant, and the difference between D-allulose synthesized at 40 and 45 °C and the optimal temperature was significant. As the temperature increases, the content of D-allulose and D-allose first increases and then decreases (Figure 8A). The results showed that the addition of 0.2 mg DPE and 0.4 mg L-RI to the whole red period of ‘Xiangfenmuzao’ as raw materials for the reaction of D-allulose and D-allose was the most obtained at 60 °C, the yield of D-allulose was 27.26%, the yield of D-allulose was 17.31%, and the yield of D-allulose: D-allose was approximately 3:2. Therefore, the optimal reaction temperature of the reaction system was determined to be 60 °C.

The yield of rare sugars synthesized by enzymatic methods is affected by several factors: temperature, pH value, substrate concentration, enzyme amount, and metal ions and their concentrations. Therefore, the ratio of DPE and L-RI double enzyme coupled enzymes should be studied to increase the yield of D-allulose and D-allose. D-allulose and D-allose were produced the most when the L-RI: DPE enzyme ratio was 10:1, with a yield of 26.13% of D-allulose and a yield of 17.52% of D-allose (Figure 8B). The synthesized products did not differ significantly when L-RI:DPE was 2:1 and 4:1, but the amount of D-allose produced gradually increased with the increase of L-RI enzyme addition, and D-allose was significantly correlated with other enzyme addition ratios when the enzyme addition ratios were 8:1 and 10:1. The amount of catalyst added affected the production of the product, and the results showed that the amount of D-allose gradually increased when the amount of L-RI enzyme was more than six times the amount of DPE enzyme. pH affects enzyme activity; the optimal pH for DPE and L-RI is known to be 7.5. pH affects both DPE and L-RI enzyme activity, and when D-allulose production decreases, D-allulose also decreases. It is most suitable to catalyze the coupling of DPE and L-RI at pH 7.5, as the production of D-allulose and D-allose is the highest, the yield of D-allulose is 28.47%, and the yield of D-allose is 16.54% (Figure 8C). The lowest (pH = 6) and highest (pH = 7.5) production of D-allulose was 17.6%, and the difference between the lowest (pH = 5.5) and highest (pH = 7.5) was 25.5%. Therefore, it is speculated that the influence of pH on the coupling of two enzymes is significant. At optimum pH, the difference between D-allose and D-allulose and other conditions was significant.

### 2.6. Sugar Conversion in the Fruit of Different Jujubes

The conversion ratio of DPE and GI double-enzyme conjugated sugars was determined using ‘Xiangfenmuzao’ juice as the substrate. After 3 h, the conversion rate of D-allulose reached a stable level, and the sugar content did not change after 7 h. The D-fructose conversion rate was 11.3%, the glucose conversion rate was 3.4%, and the D-allulose conversion rate was 12.7% (Figure 9A). When the reaction reached equilibrium, the content of D-fructose changed from 36.3% to 47.6%, and its conversion rate was 11.3%. Glucose changed from 48% to 44.6%, its conversion rate was 3.4%, and the D-allulose conversion rate was 12.7%.

To convert fructose to D-allulose, there have been studies that directly use jujube juice to add DPE to convert fructose to D-allulose, but its conversion rate has not exceeded 30%, and it is speculated that there may be substances in jujube juice that inhibit its catalysis. The fructose conversion rate of ‘Xiangfenmuzao’ jujube juice is the highest, at 30.4%. The lowest conversion rate was ‘Popozao’, at 20.7% (Figure 9B). The conversion rate of ‘Jinsixiaozao’ and ‘Zanhuangdazao’ was approximately 28%, indicating that DPE can catalyze the conversion of fructose to D-allulose, which is suitable in most jujube.

### 2.7. Catalytic Analysis of DPE and L-RI Double Enzyme Coupling in Jujube Fruit

The DPE and L-RI double enzyme coupling reaction using ‘Jinsixiaozao’, the yield of D-allulose increased with time and was 28.9% at 7 h; the yield of D-allose gradually increased and was approximately 17%; while the content of D-fructose gradually decreased with the production of D-allulose and D-allose, and the final yield was 53.6% (Figure 10A). It can be seen that the coupling reaction of DPE and L-RI was carried out by ‘Xiangfenmuzao’, and the yield of D-allulose gradually increased at 0–3 h, and the yield was 29.2%. At 0–1 h, the yield of D-allose gradually decreased, and the yield was approximately 17%; the yield of D-fructose gradually decreased, and the yield was 53% at 7 h (Figure 10B). The two-enzyme coupling conversion of two jujube juices showed that the yield of D-allulose was approximately 29% and that of D-allose was approximately 17%, indicating that this rare sugar processing process was suitable for the processing of jujube juice and had good effects.

## 3. Discussion

To optimize the reaction conditions of the double enzyme coupling system of DPE and L-RI, this study mapped the most suitable enzymatic activity factors for both enzymes, including pH, temperature, and metal ions. The results showed that the pH value of both DPE and L-RI was 7.5, with the pH value of L-RI being the most suitable pH value for L-RI from *Clostridium stercorarium* [36] and *Thermobacillus composti* KWC4 [37], indicating that pH 7.5 was the optimal pH value for L-RI. The same pH facilitates the combined reaction of the two enzymes, which can greatly reduce the nonspecific browning of the substrate and increase conversion efficiency. Temperature is an important factor affecting the enzymatic activity of DPE and L-RI. High temperature reactions and enzyme thermal stability are essential for enzyme biocatalysis during biotransformation, and high reaction temperatures increase substrate efficiency to produce conversion efficiency and reduce residue and contamination problems [38]. Different sources of DEP and L-RI may be the main reason for influencing the optimum reaction temperature, for example, DPE from *Treponema primitia* [39], *Dorea* sp. [40], and *Arthrobacter globiformis strain* [41] has an optimum reaction temperature of 70 °C. For L-RI from *T. maritima* ATCC 43589 [42] and *C. saccharolyticus* ATCC 43494 [43], the optimum reaction temperatures were 85 and 90 °C, respectively. Different types of metal ions had different effects on enzyme activity. In our study, Mn^2+^ had the strongest promotion effect on the enzyme activity of DPE and L-RI, increasing the enzyme activity by 26.5% and 38.1% compared to those without metal ions, indicating that Mn^2+^ was the most suitable metal ion for the reaction of DPE and L-RI. Enzyme activity was influenced by metal ion concentration, showing dependence on metal ion concentration.

The single DEP enzyme is less efficient at converting dilute sugars and can only convert the synthesized D-allulose from fructose. In this study, a double enzyme coupling of DEP and L-RI was used to convert the synthesized D-allulose from DEP, while L-RI consumed the fructose-converted D-allulose and further synthesized D-allulose, improving the fructose conversion rate. The highest conversion of DEP and L-RI double enzyme coupling efficiency was 46%, far exceeding that of single DEP or L-RI conversions, e.g., engineered strains of DPE *E. coli* (*Escherichia coli*) from Agrobacterium tumefaciens and DPE *Corynebacterium glutamicum* from *Flavonifractor plautii* (*Corynebacterium glutamicum*) engineered strains, using D-fructose as catalyst substrate, showed a conversion rate of approximately 30% under optimal conditions [44,45]. Temperature has a strong influence on double enzyme coupling, too high or too low temperatures can inhibit the synthesis of D-allulose and D-allose, where the conversion rate of D-allulose is related to the conversion rate of D-allose, and the conversion rate of D-allulose increases with a high conversion rate of D-allose.

Jujube juice is high in calorie sugars such as glucose, fructose, and sucrose, which can have negative effects on the body when consumed excessively. To create a low-calorie jujube juice beverage, a multistage double enzyme coupling method was used to generate D-allulose and D-allose, using jujube juice as the raw material for multienzyme catalysis. This study reported that DPE and L-RI are primarily based on monosaccharides. Double enzyme coupling of D-allose ketose and fructose (3%) yielded D-allulose at 25.2% and 8%, respectively, while double enzyme coupling of DPE and L-RI resulted in D-allulose with yields of 29% and 17%, respectively, which are higher than previously reported yields. The source of DPE and L-RI used may have contributed to this result. 

To catalyze the process, D-psicose-3-epimerase was expressed in food-grade *Bacillus subtilis*, while L-rhamnose isomerase was expressed in *Escherichia coli*. Protein inactivation was performed after the reaction, and the food-grade additive citric acid was added, as well as trace amounts of Tris-HCl and MnCl to improve the catalytic system at a later stage. 

This study conducted two double enzyme coupling, sucrose hydrolysis with GI and DPE, and a DPE and L-RI double enzyme coupling reaction, as well as a DPE single enzyme catalytic test, which can be selected according to processing needs. However, the specific use of DPE in jujube juice processing, whether GI, DPE, and L-RI can be used for simultaneous multienzyme catalysis in jujube juice, and the optimal conditions for multienzyme catalysis still need to be investigated. Further research can be conducted in this direction to form a more complete system for the processing of jujube juice with double diluted sugar, which simplifies processing into three steps: water bath heating to obtain sugar-extracted jujube juice, sucrose hydrolysis, and multienzyme catalysis of GI, DPE, and L-RI to obtain jujube juice containing D-allulose and D-allose.

## 4. Materials and Methods

### 4.1. Materials

In 2021, samples were collected from Cangxian National Jujube Resource Garden in Hebei Province, and jujube fruits of 8 varieties were collected from ‘Jinsixiaozao’, ‘Xiangfenmuzao’, ‘Huizao’, ‘Zanhuangdazao’, ‘Popozao’, ‘Linyilizao’, ‘Yazao’, and ‘Dasuanzao’ in a total of 5 stages, including the young fruit stage, fruit expansion stage, white-mature stage, half-red stage, and full-red stage, and the samples were ground into powder and stored at −80 °C.

D-allulose and D-allose standards were purchased from Shanghai Yuanye Biotechnology Co., Ltd. (Shanghai, China). Agarose gel, acetonitrile, purified water, and an affinity chromatography column were purchased from Kangwei Century Biotechnology Co., Ltd. (Taizhou, China). Glucose, fructose, and sucrose standards were purchased from Beijing Solabal Technology Development Co., Ltd. (Beijing, China). All the other chemicals used in this study were of analytical grade and used without further purification. Double-distilled water was used in all experiments.

### 4.2. Glucose, Fructose, and Total Sugar Detection

Preparation of standard solution: glucose, D-allulose, fructose, D-allose, and sucrose standards were prepared at a mass concentration of 100 mg/mL, each containing 1 mL added to a 50 mL volumetric flask to set the volume, 2 mg/mL mixed standard solution containing 5 sugars, and the same method prepared a mixed standard solution at a mass concentration of 0.5, 1.5, 2, and 2.6 mg/mL.

Detection method: The high-performance liquid chromatography (HPLC) detection conditions: mobile phase, 75:25 acetonitrile/water (*V*/*V*); flow rate, 0.7 mL/min; injection volume. 10 μL, column temperature, 35 °C, analytical column, NH2P-50 4E model chromatography column; and detector, evaporative light detector (ELSD). Sugars were detected by HPLC.

Total sugar detection method: A 3,5-dinitrosalicylic acid colorimetric method was used to detect the total sugar of jujube fruit.

Sucrose hydrolysis: 10 mL of jujube juice was added to 0.5% citric acid in a water bath at 80 °C for 4 h, and HPLC was used to detect the hydrolysis of sucrose in jujube juice.

### 4.3. Recombinant Plasmid Construction and Enzyme Expression and Purification

The DPE and L-RI nucleotide sequences queried by NCBI were synthesized by General Biosystems (Anhui) Co., Ltd. (Chuzhou, China), and the DPE, NdeI, and Xhol L-RI genes were recombined into pMA5 and pET-22b (+) vectors using NdeI-BamH, converted into *B. subtilis* WB600, *E. coli* BL21 (DE3) competent states, coated in LB-Kana (50 μg/mL) and LB-Amp (100 μg/mL) solid medium, and validated by PCR.

Recombinant *B. subtilis* WB600 carrying the DPE gene and recombinant *E. coli* BL21 (DE3) carrying the DPE gene were cultured overnight in LB-Kana (50 μg/mL) medium and LB-Amp (100 μg/mL) medium, respectively. After the end of the culture, the cultures were centrifuged for 10 min (4 °C, 6000 r/min), the supernatant was discarded, the culture was resuspended with 30 mL of lysate (25 mM Tris-HCl, 300 mM NaCl, and 40 mM imidazole, pH 8.0), and the cells were ruptured by ultrasonication on ice. The mixture was centrifuged at 4 °C and 6000 r/min for 30 min, and the supernatant was collected as the crude enzyme solution.

The crude enzyme solution was purified by Ni-NTA affinity column low-concentration imidazole eluate to elute heteroproteins and high-concentration imidazole-eluting target proteins.

### 4.4. Enzyme Assays 

One DPE enzyme activity unit is defined as the amount of enzyme required to produce 1 μmol of D-allulose per minute when reacted at 50 °C at pH = 7.5. The catalytic system consisted of 0.5 mL reaction system, 80 mg/mL D-fructose, and 160 mM HEPES buffer with 0.3 mg DPE pure solution, and the mixture was allowed to react for 30 min prior to inactivation and HPLC method analysis to determine DPE enzyme activity.

One L-RI enzyme activity unit is defined as the amount of enzyme required to produce 1 μmol of D-allulose per minute when reacted at pH = 7.5 at 65 °C. The catalytic conditions included a 0.5 mL reaction system, 100 mg/mL D-allulose added to 0.2 mg L-RI pure solution, and the other steps were the same as above.

### 4.5. Effect of Temperature, pH and Metal Ions on Enzymatic Reactions

In a 0.5 mL reaction system, 0.2 g DPE, 20 mM HEPES (pH = 8.0), and 50 mM D-fructose were added for 30 min. The reaction temperature was 20–70 °C, and the inactivation was 10 min. In the 0.25 mL reaction system, 0.1 mg L-RI, 50 mM Tris (pH 7.5), and 50 mM D-allulose were added at a reaction temperature of 40–80 °C and inactivated. HPLC was used to detect enzyme activity and determine the optimal enzymatic reaction temperature for DPE and L-RI.

In a 0.5 mL reaction system, 0.2 mg DPE, 20 mM HEPES (pH = 8.0), and 50 mM D-fructose were added, and the reaction was performed under the optimal conditions of the above test for 30 min. The reaction pH was 6.5–9.5, and the buffer included MES buffer (pH 5.5–7) and HEPES buffer (pH 7.5–9.5). In the 0.25 mL reaction system, 0.1 mg L-RI, 50 mM Tris, and 50 mM D-allulose were added and reacted under optimal conditions of the above test for 30 min. The reaction pH was 4.5–8.5, and acetic acid–sodium acetate buffer (pH 4–7) and PBS buffer (pH 7–7.5) were used. HPLC was used to detect enzyme activity and determine the optimal pH of DPE and L-RI enzymatic reactions.

In the 0.5 mL reaction system, 1 mM Mn^2+^, Mg^2+^, Cu^2+^, Co^2+^, Ni^2+^, Ba^2+^, and Ga^2+^ metal ions, 0.2 mg DPE, 20 mM HEPES, and 50 mM D-fructose were added to the above test under optimal conditions for 30 min. In a 0.25 mL reaction system, 1 mM of different metal ions, 20 mg L-RI, 50 mM Tris, and 50 mM D-allulose were added to the reaction under optimal conditions of the above test for 30 min. HPLC was used to detect enzyme activity.

### 4.6. Effect of Mn^2+^ Concentration of the Optimal Metal Ion on Enzymatic Reactions

In the 0.5 mL reaction system, 0.5, 1, 2, 3, and 5 mM Mn^2+^, 0.2 mg DPE, 20 mM HEPES, and 50 mM D-fructose were added and reacted under the above optimal conditions for 30 min. In the 0.25 mL reaction system, different concentrations of Mn^2+^, 0.1 mg L-RI, 50 mM Tris, and 50 mM D-allulose were added for 30 min.

### 4.7. Kinetic Characterization

Michaelis–Menten fitting was performed using GraphPad Prism 8 software to analyse the parameters km and Vmax of DPE enzyme and L-RI enzyme reactions. The reaction system used the optimal conditions obtained from the above DPE enzyme and L-RI enzyme activity tests to detect the dynamic and mechanical parameters km and Vmax values of D-allulose at different concentrations. The concentration of D-allulose was set at 0.5–50 mM.

### 4.8. Effects of Temperature, Enzyme Ratio, and pH on DPE and L-RI Double Enzyme Coupling

The effects of pH, DPE and L-RI enzyme ratio, temperature, and conversion time on the sucrose conversion rate, D-allose conversion rate, and the ratio of D-allulose conversion rate to total sugar in jujube juice were studied by single factor.

To the 0.5 mL reaction system, 0.2 mg DPE, 0.4 mg L-RI, 20 mM Tris, 1 mM Mn^2+^, and 0.25 mL ‘Jinsixiaozao’ or ‘Xiangfenmuzao’ Full-red stage jujube juice were added, pH 7.5, temperature between 40~75 °C, reaction 5 h, inactivation, HPLC to detect D-allulose and D-allose production.

Indeed, 0.2 mg DPE, 20 mM Tris, 1 mM Mn^2+^, 0.5 mL jujube juice were added to the 1 mL reaction system, the ratio of L-RI to DPE was 2:1, 4:1, 6:1, 8:1, 10:1, pH was 7.5, and the HPLC detection product was detected by HPLC under the most suitable conditions of the above test for 5 h.

To the 0.5 mL reaction system, 0.2 mg DPE, 20 mM Tris, 1 mM Mn^2+^, and 0.25 mL jujube juice were added, pH value between 5.5 and 8.5, under the above optimal conditions of water bath for 5 h, HPLC was used to detect product.

### 4.9. Conversion of Glucose, Fructose, and D-Allulose in Jujube Juice

‘Xiangfenmuzao’ jujube juice hydrolyzed by 5 mL sucrose was added with 3 mg DPE and 83 U GI to react overnight at 55 °C. The changes of glucose, fructose, and D-allulose in jujube juice were detected by HPLC.0.25 mL of 8 varieties of sucrose hydrolyzed jujube juice was added to 0.2 mg DPE, 20 mM Tris-HCl, 3 mM Mn^2+^, and water was added to supplement the reaction system to 0.5 mL, and the reaction was reacted at 50 °C for 6 h to detect the fructose and D-allulose content in jujube juice.

The sucrose hydrolysis of ‘Xiangfenmuzao’ jujube and golden silk jujube juice in the full-red stage was used, and 0.5 mL of jujube juice, 20 mM Tris, and 1 mM Mn^2+^ were added to the reaction system in 1 mL, and the contents of fructose, D-allulose, and D-allose in ‘Xiangfenmuzao’ jujube and golden silk jujube juice were detected for 7 h under the above optimal reaction conditions.

### 4.10. Conversion of Glucose, Fructose, D-Allulose in Jujube Juice

All data in this paper were collected in triplicate and expressed as mean ± SD, using SPSS software (25.0 version; IBM, Inc., Chicago, IL, USA). One-way analysis of variance (ANOVA) and Duncan’s multiple comparison test were used to analyze the differences in the contents of monosaccharides and total sugars in different varieties and fruit development stages (Figure 1). Differences in temperature, enzyme addition, and pH value of the double enzyme coupling conditions were analyzed (Figure 8). In all analyses, *p* < 0.05 indicated statistical significance. The chart is drawn by Origin software.

## Figures and Tables

**Figure 1 plants-12-03084-f001:**
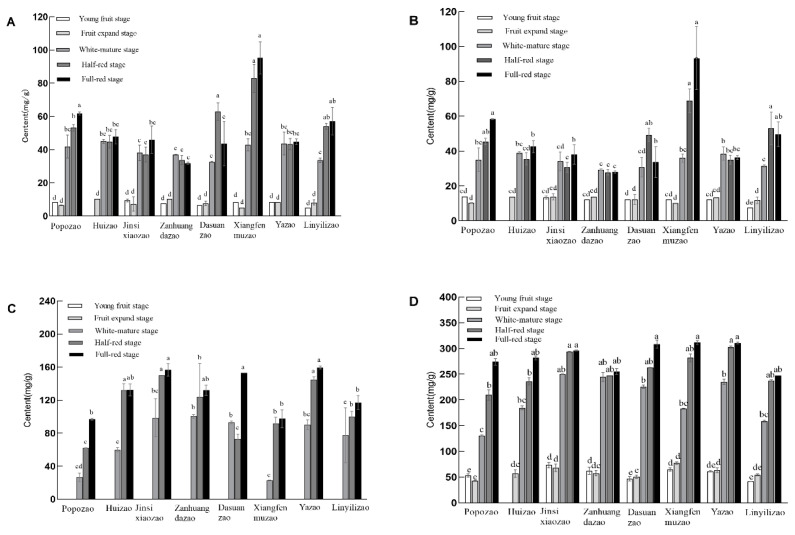
Analysis of sugar composition of jujube fruits of different varieties in different periods (mg/g). Note: (**A**): Glucose; (**B**): Fructose; (**C**): Sucrose; (**D**): Total sugar; the different small letters represent significance at the 5% level.

**Figure 2 plants-12-03084-f002:**
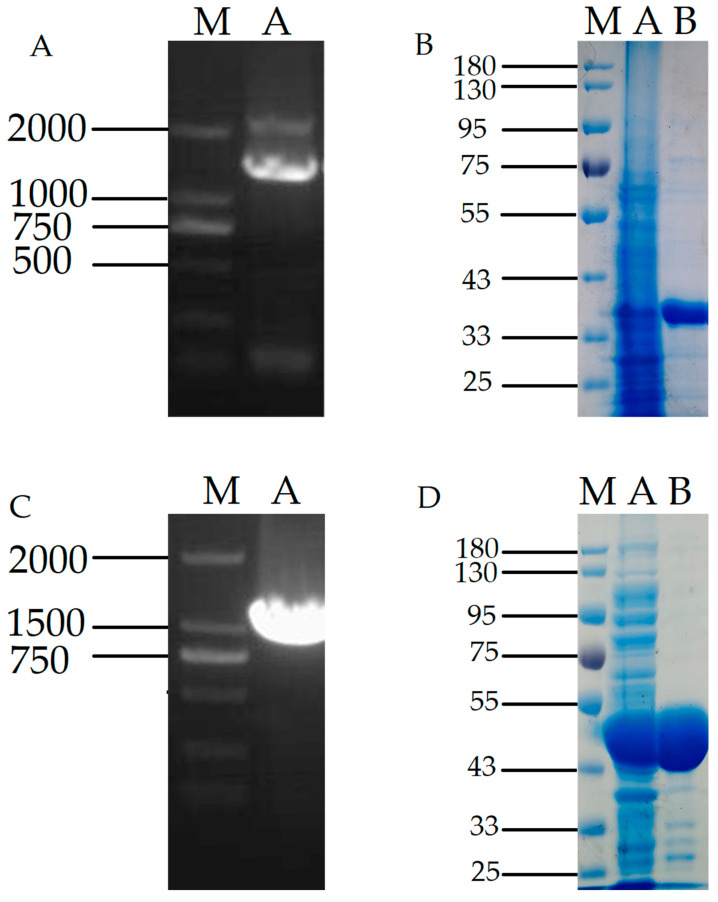
(**A**), PCR map of recombinant plasmid DPE expressed in WB600(bp); (**B**), SDS-PAGE diagram of purified product of recombinant protein DPE(kDa); (**C**), PCR map of recombinant plasmid L-RI expressed in BL21(bp); (**D**), SDS-PAGE diagram of purified product of recombinant protein L-RI (kDa). Note: (**A**), Lane M: DNA Marker; Lane A: PCR product; (**B**), M: DNA Marker; Lane A: Unpurified; Lane B: No purified product of target protein; Lane C: Containing purified product of target protein; (**C**), Lane M: DNA Marker; Lane A: PCR product; (**D**), M: DNA Marker; Lane A: Unpurified; Lane B: No purified product of target protein.

**Figure 3 plants-12-03084-f003:**
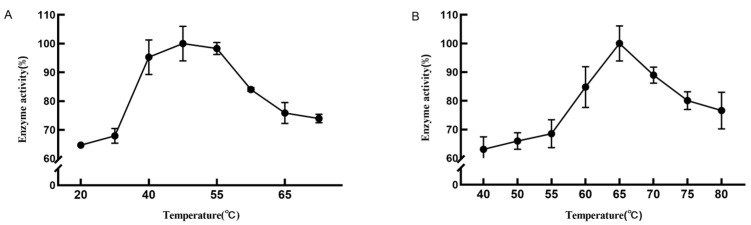
Effect of temperature on the activity of D-psicose-3-epimerase (**A**) and L-rhamnose isomerase (**B**).

**Figure 4 plants-12-03084-f004:**
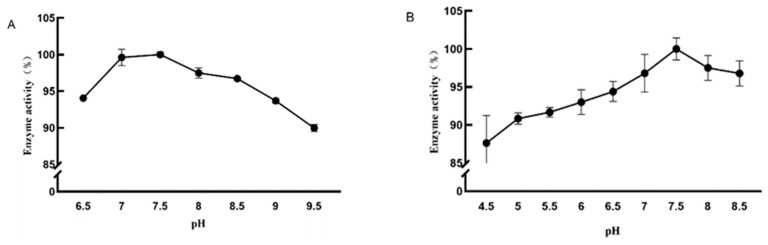
Effect of pH on the activity of D-psicose-3-epimerase (**A**) and L-rhamnose isomerase (**B**).

**Figure 5 plants-12-03084-f005:**
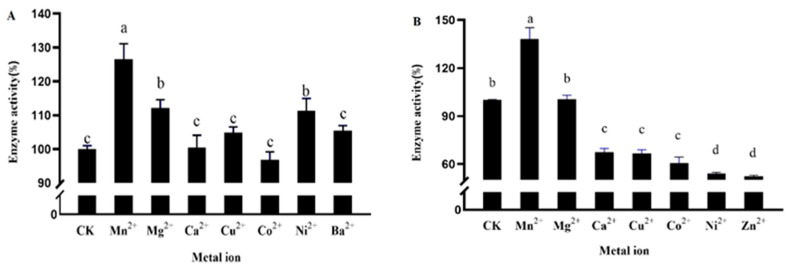
Effect of metal ions on the activity of D-psicose-3-epimerase (**A**) and L-rhamnose isomerase (**B**). Note: The different small letters represent significance at the 5% level.

**Figure 6 plants-12-03084-f006:**
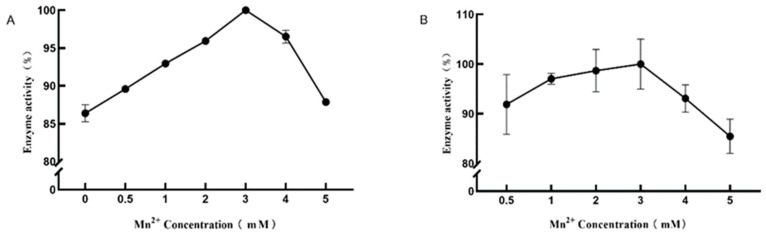
Effect of Mn^2+^ concentration on the activity of D-psicose-3-epimerase (**A**) and L-rhamnose isomerase (**B**).

**Figure 7 plants-12-03084-f007:**
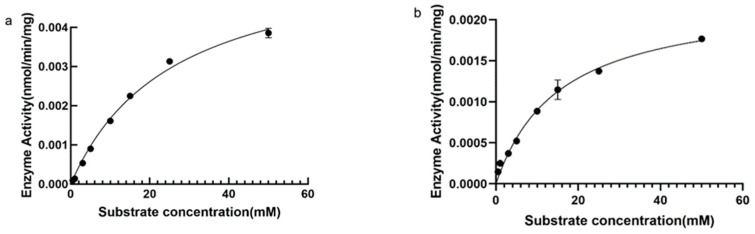
(**a**), Hyperbola of the effect of substrate concentration on DPE enzyme reaction rate; (**b**), Hyperbola of the effect of substrate concentration on L-RI enzyme reaction rate.

**Figure 8 plants-12-03084-f008:**
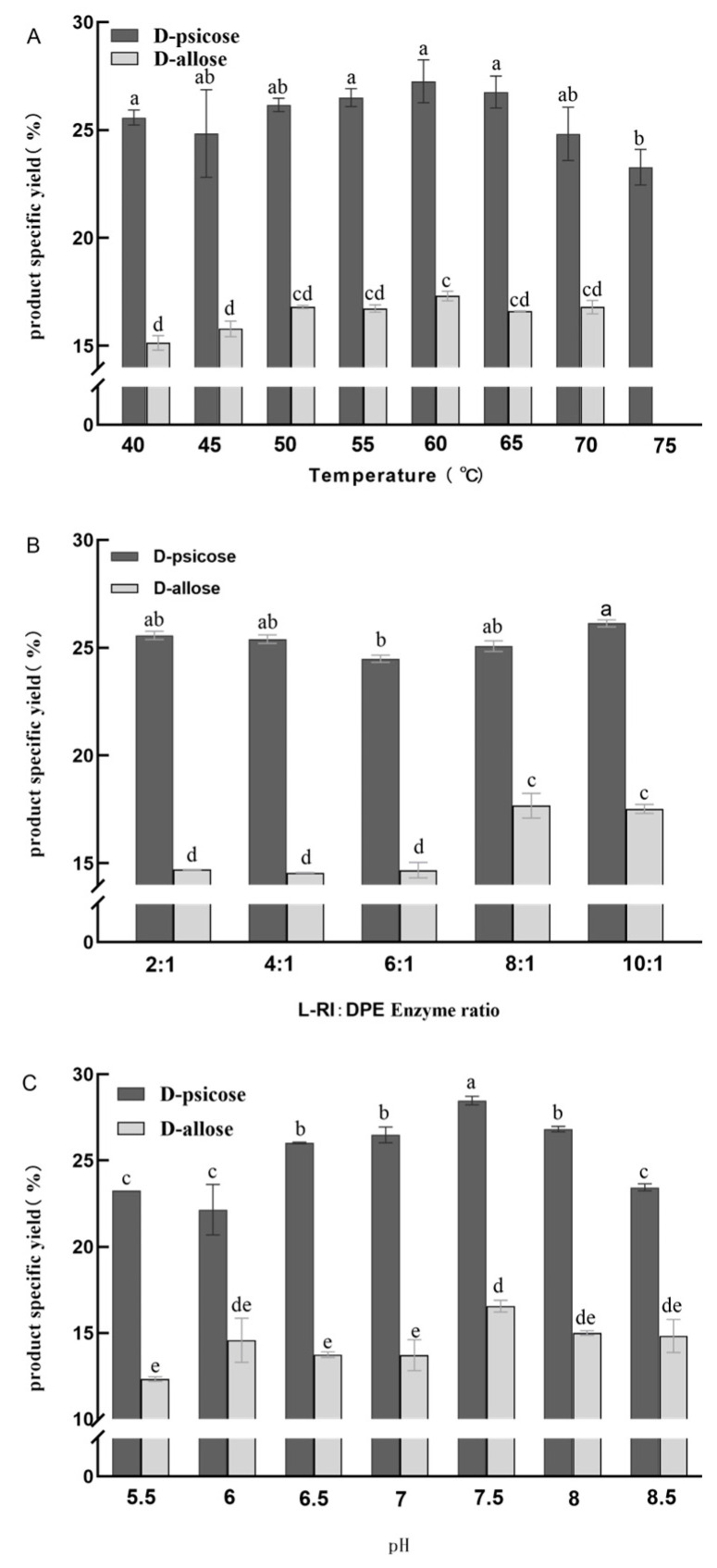
(**A**), Effect of temperature on the coupling of DPE and L-RI; (**B**), Effect of DPE and L-RI plus enzyme ratio on double enzyme coupling; (**C**) Effect of pH on the coupling of DPE and L-RI. Note: The different small letters represent significance at the 5% level.

**Figure 9 plants-12-03084-f009:**
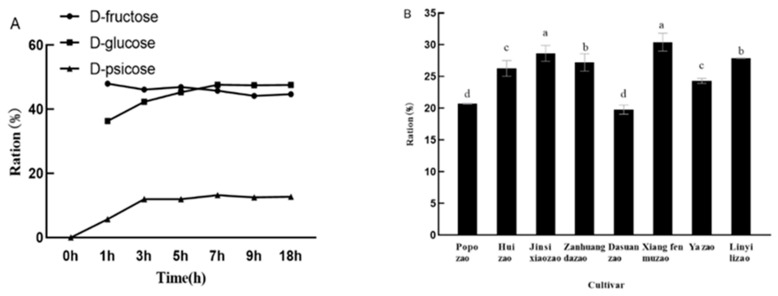
(**A**), Analysis of the conversion ratio of GI and DPE double enzyme coupling sugar; (**B**), Analysis of DPE catalyzed fructose conversion percentage. Note: The different small letters represent significance at the 5% level.

**Figure 10 plants-12-03084-f010:**
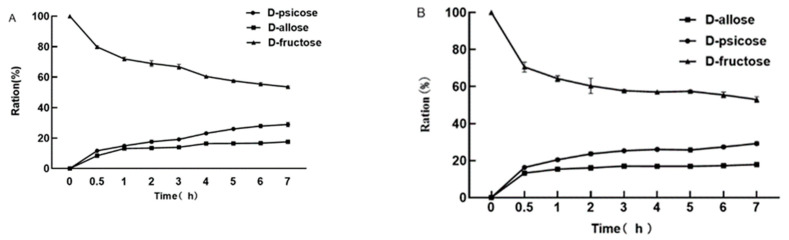
Analysis of the double enzyme coupling reaction of DPE and L-RI. Note: (**A**): Analysis of the double enzyme coupling reaction of DPE and L-RI in ‘Jinsixiaozao’; (**B**): Analysis of the double enzyme coupling reaction of DPE and L-RI in ‘Xiangfenmuzao’.

## Data Availability

Data will be made available upon request.

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
