# Peer review of "Establishment of the Biotransformation of D-Allulose and D-Allose Systems in Full-Red Jujube Monosaccharides"

_plants, 2023, doi:10.3390/plants12173084_

Round 1

Reviewer 1 Report

The manuscript 'Establishment of the biotransformation of D-allulose and D allose systems in red jujube monosaccharides (plants-2507558) describes the possibility of converting monosaccharides from jujube fruits into low-calorie natural rare sugars.

Overall, I found the manuscript well written and much of it is well described.
Nevertheless, there are many important irregularities in the text.

Title:

It is not clear what '.... in red jujube...' means? Did you use the red fruit types of jujube or did you use red stage fruits? In addition, you tested the composition of the sugar in two red stages. Which one did you use for further analysis, the half red stage or the full red stage?

Abstract:

The reader does not learn that the authors checked the suitability of the varieties and the stages of fruit development. Nor what they used for the further chemical analyses, which are explained in full detail.

Introduction:
The objective of the paper is missing.

Methods:
The description of the statistical analysis is missing.
It seems that One-Way ANOVA was used in Fig. 1 and Fig. 8, but the analysis should include 2 factors: variety and fruit stage (Fig. 1), variety and enzyme (Fig. 8).

Methods: some important data or descriptions of the work are missing. The reader can get the information about the initial work from the Results, but later it is not clear which variety, which plant material, from which stage was used for chemical analysis…

Reviewer 2 Report

In this study, the method of double enzyme coupling ( D-psicose 3-epimerase and l-rhamnose isomerase ) was used to transform glucose and fructose of jujube juice to synthesize D-allulose and D-allose, which significantly improved the conversion rate of D-allulose and D-allose. At the same time, the enzyme activity conditions of DPE and L-RI were optimized, and the optimum enzyme activity conditions were screened. The method of double enzyme coupling was used to biotransform rare sugars from jujube juice, which enriched the nutritional properties of jujube juice and improved the economic value of jujube juice. In addition, there are also the following problems.

1.Abstract : The authors should discuss the importance of converting rare sugars from the perspective of jujube juice nutrition, and how the conversion of rare sugars from jujube juice plays a role in the development of new products and functional beverages. What are the benefits of D-allulose and D-allose to the human body and what are the applications in real life ?

2. The article lacks the latest papers on the composition of jujube sugar, proves that glucose, fructose and sucrose are the main sugar components of jujube juice, and discusses the composition and distribution of sugar in jujube juice.

3. The catalytic conditions of a single enzyme were studied, including the effects of temperature, pH and metal ions on D-psicose 3-epimerase and l-rhamnose isomerase. Paragraph 3.5 lacks the study of the combined catalysis of metal ions on the two enzymes, please explain the reason.

4. A variety of studies have shown that Co can increase the enzyme activity of D-psicose 3-epimerase and l-rhamnose isomerase. In this study, the activity of the two enzymes was inhibited. Please analyze the reasons for the inhibition.

5. Please unify the title and the writing of D-allulose and D-psicose in the text.

6. Please unify the text size and image size of Figure 2.The picture fonts in the text should be unified, and the fonts in Figure 8 are not clear enough.

7. Technical mistake are many and should be re-checked as well.

None

Round 2

Reviewer 1 Report

The revised version of the manuscript 'Establishment of the biotransformation of D-allulose and D allose systems in red jujube monosaccharides (plants-2507558) was a kind of improvement of the original manuscript. I still have concerns about some points I mentioned in my previous review.

Title:

The authors explained the title of the paper in detail, but did not change it. I think that only changing the title will prevent future inconsistencies or that the title will not raise questions, as it did for me.

Summary:

What purpose does the sentence inserted at the beginning serve? This sentence explains nothing, it only confuses. Extra: The phrase 'in jujube juice' is repeated three times. Also, in this version, the words 'in this study' are repeated at the very beginning of the sentence in the first two sentences. In response to comments, the authors wrote a statement that they had corrected the abstract, but these changes are not found in the abstract.

Introduction:

The objective of the paper is still missing.

The authors added a couple of sentences to the first paragraph that do not belong in the introduction.
